# The Relationship Between Kidney Biomarkers, Inflammation, Severity, and Mortality Due to COVID-19—A Two-Timepoint Study

**DOI:** 10.3390/ijms26136086

**Published:** 2025-06-25

**Authors:** Sara Soares Tozoni, Ana Carolina Gadotti, Erika Sousa Dias, Julia Bacarin Monte Alegre, Beatriz Akemi Von Spitzenbergen, Marina de Castro Deus, Thyago Proença de Moraes, Andrea Novais Moreno-Amaral

**Affiliations:** Pós-Graduação em Ciências da Saúde, Pontifícia Universidade Católica do Paraná (PUCPR), Curitiba 80215-901, Brazil; saratozoni@gmail.com (S.S.T.); ana_raixu3@hotmail.com (A.C.G.); erika.sousa@pucpr.edu.br (E.S.D.); julia.alegre@pucpr.edu.br (J.B.M.A.); beatriz.van@pucpr.edu.br (B.A.V.S.); marina.by@hotmail.com (M.d.C.D.); thyago.moraes@pucpr.br (T.P.d.M.)

**Keywords:** COVID-19, inflammation, kidney disease, Interleukin-6, mortality, cytokines

## Abstract

About a quarter of COVID-19 patients develop acute kidney injury (AKI), worsening prognosis and increasing mortality. Severe COVID-19 often triggers a hyperactive immune response, influencing disease outcomes. This study examined the correlation between kidney injury biomarkers, inflammatory mediators, and mortality in COVID-19 patients. Blood samples from 390 COVID-19 patients were collected at admission and before the outcome. Serum Cystatin C (CysC), albumin, and plasma NGAL were measured via nephelometry, while inflammatory mediators (IL-4, IL-6, IL-10, IL-15, IFN-γ, TNF-α, and IL-1β) were assessed by ELISA. Most patients were male, with hypertension and diabetes as common comorbidities, and a high ICU admission rate. Lower albumin and elevated CysC and NGAL were linked to mortality. Increased inflammatory mediators correlated with lower albumin and higher CysC and NGAL, reinforcing the connection between systemic inflammation and kidney dysfunction. Elevated cytokines and kidney injury biomarkers, including NGAL, CysC, and low albumin, are strongly associated with higher mortality in COVID-19 patients. These findings highlight the role of inflammation and kidney function markers in identifying high-risk individuals, improving patient management, and mitigating complications. Monitoring these biomarkers remains crucial for managing long-term health impacts and future outbreaks

## 1. Introduction

COVID-19 has been associated with an increased risk of acute kidney injury (AKI) [1], which in turn elevates mortality rates [2]. The SARS-CoV-2 virus directly interacts with the angiotensin-converting enzyme 2 (ACE-2) receptor, highly expressed in kidney tubular cells [3], causing cellular damage, triggering inflammatory responses, and impairing kidney function [4]. In severe cases of COVID-19, AKI is a key contributor to increased morbidity and mortality. In the U.S., over 60% of patients needing renal replacement therapy (RRT) developed AKI [5]. In Brazil, AKI incidence and outcomes varied by healthcare setting. A public hospital in São Paulo reported a 77% AKI incidence in ICU patients, with 65.4% mortality [6]. In a private Rio de Janeiro hospital, AKI affected 55.9% of ICU patients, with a mortality rate of 33.3% versus 8.9% in those without AKI [7].

Multiple factors may contribute to AKI in COVID-19, including direct viral cytopathic effects, cytokine-driven inflammation, microvascular injury, and immune-mediated damage [8]. Even post-pandemic, biomarkers collected at admission or during hospitalization remain valuable for identifying patients at increased risk of severe outcomes, as virus–host interactions persist. A growing body of evidence has demonstrated that serum cystatin C (CysC), albumin, and plasma/urine neutrophil gelatinase-associated lipocalin (NGAL) are valuable early diagnostic biomarkers for AKI [9,10,11]. These biomarkers have also been shown to strongly predict worse outcomes in COVID-19 patients [12,13,14,15]. In parallel, the pandemic has underscored the role of a dysregulated immune response, known as cytokine release syndrome (CRS), characterized by the overproduction of inflammatory cytokines like TNF-α, IL-6, IL-1β, and IFN-γ, which serve as inflammatory biomarkers of COVID-19. CRS has been linked to the development of multiple organ dysfunctions, including widespread endothelial dysfunction, microvascular thrombosis, and organ damage, particularly affecting the kidneys [8]. Among the prospectively assessed markers, IFN-γ, IL-6, IL-1β, and IL-4 emerged as significant predictors of death, with IL-4 and IL-1β demonstrating the highest predictive performance for distinguishing survivors from non-survivors [16].

Accurate biomarkers are essential for early prognosis prediction, allowing timely preventive interventions and optimized management for high-risk patients. Understanding these biomarkers can help clarify the complex interplay between the host’s immune response and renal injury in COVID-19, potentially preventing complications such as AKI, which significantly impact disease outcomes and increase mortality risk. This study aimed to investigate the correlation between kidney biomarkers and inflammatory mediators, as well as their association with mortality risk in individuals diagnosed with COVID-19.

## 2. Results

### 2.1. Study Population

The mean age of the 390 patients was 54.6 ± 16.4 years, with 62.3% being male (Table 1). The participants’ estimated glomerular filtration rate (eGFR) and body mass index (BMI) were 76 ± 30 mL/min/1.73 m^2^ and 29.9 ± 4.9, respectively. Systemic arterial hypertension (SAH) and diabetes mellitus (DM) were the most common conditions in this population, affecting 39.5% and 22.8% of patients, respectively. The demographic and clinical characteristics of the population are presented in Table 1. Participants were classified as having mild or severe COVID-19 based on disease severity [17]. Of the 384 participants classified, 67.2% were in the severe group (Table 1). 

The association between clinical and demographic variables with death was assessed using logistic regression. Individuals older than 65 years had a 9-fold higher risk of death compared to younger individuals (OR 8.97. 95% CI: 4.88–16.48. *p* < 0.001) (Table 2). Those with cardio-metabolic disease also had an increased risk of death (OR 4.8; 95% CI: 2.41–9.59; *p* < 0.001). Patients with more than four comorbidities and fewer than seven days of symptoms before admission had a 5-fold and nearly 3-fold higher risk of death, respectively (OR 5.1. 95% CI: 2.79–9.33. *p* < 0.001; OR 2.65. 95% CI: 1.48–4.73. *p* < 0.001) (Table 2).

### 2.2. Kidney Biomarkers Levels

Albumin, CysC, and NGAL levels were measured at two time points during the patient’s hospitalization. The first blood sample was collected upon study admission (admission sample), and the last was obtained before the outcome, which could be either discharge or death (outcome sample).

In the discharge group, the albumin level in the admission sample (n = 304) was 2668 ± 569 mg/dL, while in the outcome sample (n = 163) it was 2545 ± 759 mg/dL. In the death group, albumin levels were 2019 ± 581 mg/dL in the admission sample (n = 55) and 1549 ± 97 mg/dL in the outcome sample (n = 44) (Table 3). A comparison of albumin levels between discharge and death groups revealed that the death group had significantly lower albumin levels than the discharge group in the admission sample (*p* = 0.003). Similarly, albumin levels in the outcome sample were lower in the death group compared to the discharge group (*p* = 0.001). Albumin levels were significantly lower at the outcome time point in the death group compared to admission (*p* = 0.001) (Figure 1A). The time points of the biomarker levels assessed in both groups were associated with mortality in the logistic regression models (Table 3).

The CysC level in the discharge group’s admission sample (n = 304) was 1.04 ± 0.38 mg/L, and in the outcome sample (n = 163) it was 0.96 ± 0.45 mg/L (Figure 1B). In the death group, the mean CysC level in the admission sample (n = 55) was 1.86 ± 1.48 mg/L, and in the outcome sample (n = 44), it was 2.22 ± 1.14 mg/L (Table 3). CysC levels in the death group were higher than in the discharge group at both time points: admission sample (*p* = 0.001) and outcome sample (*p* = 0.001). Additionally, comparison of CysC levels between the two time points showed that the discharge group had significantly lower levels in the outcome sample (*p* = 0.001) (Figure 1B). CysC levels were associated with mortality at both time points in both groups (Table 3). 

The NGAL level in the discharge group was 118.9 ± 103.1 ng/mL in the admission sample (n = 196), and 133.2 ± 137 ng/mL in the outcome sample (n = 31). In the death group, NGAL levels were 247 ± 236.2 ng/mL in the admission sample (n = 44) and 1121 ± 1217 ng/mL in the outcome sample (n = 8) (Table 3). NGAL levels were significantly higher in the death group than the discharge group at both time points: admission sample (*p* = 0.001) and outcome sample (*p* = 0.004). Additionally, NGAL levels were significantly higher in the death group when comparing the two time points (*p* = 0.005) (Figure 1C). However, it is essential to note that NGAL was assessed in a limited number of outcome samples (only 31 in the discharge group and 8 in the death group), which limits the statistical power and warrants cautious interpretation of these findings. Although NGAL was correlated with mortality at both time points, this association was not statistically significant in the multivariable model at outcome, likely due to reduced sample size (Table 3).

### 2.3. Correlation Between Kidney Biomarkers and Inflammatory Mediators

The correlation between inflammatory mediators and biomarkers was assessed using only the first blood sample from each patient. ELISA was used to evaluate inflammatory mediators, while albumin, CysC, and NGAL levels were measured through nephelometry. Albumin showed an inverse correlation with IL-6 (*p* < 0.001), IFN-γ (*p* < 0.001), IL-4 (*p* < 0.001), TNF-α (*p* < 0.001), IL-15 (*p* = 0.016), and IL-1β (*p* < 0.001) (Figure 2A–F).

CysC exhibited a direct correlation with several cytokines, indicating that as CysC levels increased, so did the levels of IL-4 (*p* = 0.002), TNF-α (*p* = 0.006), IL-15 (*p* = 0.012), and IL-1β (*p* = 0.009) (Figure 3A–D). NGAL showed significant positive correlations with IL-6 (*p* = 0.003), IFN-γ (*p* = 0.023), IL-4 (*p* = 0.002), and IL-15 (*p* = 0.016) (Figure 4A–D), indicating that as NGAL levels increased, so did the levels of these cytokines.

### 2.4. Mortality Risk Correlated with Kidney Biomarkers and Inflammatory Mediators

Admission samples were analyzed to evaluate how inflammatory mediators and kidney biomarkers influence mortality risk in the studied population. The mortality risk increased notably; it was particularly associated with the lowest albumin concentration across all cytokines. IL-6 adjusted for albumin was associated with death (*OR*: 1.15. 95% CI: 1.06–1.24. *p* < 0.001) (Figure 5A). Similarly, IL-10 was associated with death (OR: 1.07. 95% CI: 1.02–1.11. *p* = 0.001); IFN-γ (OR: 1.12. 95% CI: 0.99–1.26. *p* = 0.05); IL-4 (OR: 5.66. 95% CI: 2.43–13.17. *p* < 0.001); TNF-α (OR: 1.34. 95% CI: 1.08–1.65. *p* = 0.006), IL-15 (OR: 2.04. 95% CI: 1.55–2.68. *p* < 0.001); and IL-1β (OR: 2.04. 95% CI: 1.55–2.68. *p* < 0.001) (Figure 5B–G) were all significantly associated with death.

Referring to inflammatory mediators adjusted for CysC, the mortality risk increased primarily with higher concentrations of CysC. This was observed for IL-6 (OR: 1.25. 95% CI: 1.14–1.37. *p* < 0.001); IL-10 (OR: 1.05. 95% CI: 1.01–1.09. *p* = 0.008); IFN-γ (OR: 1.14. 95% CI: 1.02–1.26. *p* = 0.01); IL-4 (OR: 6.12. 95% CI: 2.38–15.7. *p* < 0.001); TNF-α (OR: 1.37. 95% CI: 1.20–1.56. *p* < 0.001), IL-15 (OR: 1.87. 95% CI: 1.46–2.40. *p* < 0.001) and IL-1β (OR: 3.10. 95% CI: 1.91–5.04. *p* < 0.001) (Figure 6A–G).

Furthermore, the mortality risk was higher at elevated NGAL concentrations. IL-6 adjusted for NGAL showed an increased risk of death (OR: 1.16. 95% CI: 1.06–1.26. *p* < 0.001) (Figure 6A). Similarly, IL-10 (OR: 1.20. 95% CI: 1.03–1.40. *p* = 0.01); IL-4 (OR: 5.69. 95% CI: 1.99–16.29. *p* = 0.001); IL-15 (OR: 1.89. 95% CI: 1.43–2.49. *p* < 0.001) and IL-1β (OR: 2.52. 95% CI: 1.72–3.70. *p* < 0.001) were also associated with increased mortality risk (Figure 7A–E).

## 3. Discussion

In this cohort study of hospitalized patients with COVID-19, elevated levels of the kidney biomarkers CysC and NGAL along with lower albumin concentrations were significantly different between patients who were discharged and those who died. The study found a strong correlation between markers of renal injury (CysC and NGAL) and inflammatory mediators (albumin, IL-6, IFN-γ, IL-4, TNF-α, IL-15, and IL-1β), underscoring the impact of CRS involvement on renal injury during SARS-CoV-2 infection. The correlation between renal injury markers and cytokines indicated an increased risk of death in the study population, emphasizing the potential value of these biomarkers for future clinical scoring systems and prediction models. Associations between low albumin and high cystatin C levels with increased mortality have been described in previous studies [18,19]. Our findings confirm and reinforce these associations in a distinct population of hospitalized COVID-19 patients in Brazil, contributing to the existing evidence by validating the prognostic relevance of these biomarkers in a geographically and clinically diverse population.

Although the WHO declared an end to the Public Health Emergency of International Concern in May 2023, COVID-19 persists, with ongoing hospitalizations and deaths, particularly among vulnerable and unvaccinated individuals in low-income countries [20]. Emerging variants with higher transmissibility and severity continue to pose risks, especially for those with preexisting conditions. Our cohort reflects typical COVID-19 demographics: age over 65, high prevalence of cardio-metabolic diseases (52.2%), hypertension (39.5%), diabetes (22.8%), multiple comorbidities, and admission SpO2 below 95%. Many patients required immediate ICU care. Comorbidities like hypertension and diabetes, known risk factors for chronic kidney disease, increase susceptibility to severe outcomes, including kidney complications [21], by exacerbating inflammation, CRS, and multi-organ dysfunction. These comorbidities, known as risk factors for chronic kidney disease, also amplify the inflammatory response in COVID-19, contributing to CRS and multi-organ dysfunction. Notably, most of the cohort was enrolled before widespread vaccine availability in Brazil, with data collection ending in August 2021, reflecting a predominantly unvaccinated or partially vaccinated population.

COVID-19 severity is linked to elevated levels of inflammatory mediators, including cytokines (e.g., IL-2, IL-7, IL-10, IFN-γ, and TNF-α) and chemokines (e.g., G-CSF, MCP1, MIP1α, and CXCL10), as well as markers like C-reactive protein, ferritin, and D-dimers [22,23,24,25,26]. Fatal COVID-19 cases are often associated with CRS driven by a cytokine storm [27,28,29]. Our findings align with these observations, showing elevated cytokine levels from admission to outcome. In COVID-19, elevated TNF-α and IL-6 levels may suppress albumin synthesis and contribute to hypoalbuminemia. Additionally, increased vascular permeability and capillary leakage—common in acute inflammatory conditions—can redistribute albumin from the serum to the interstitial space, reducing serum albumin levels [30,31]. Our results revealed a significant inverse correlation between lower albumin levels and higher concentrations of cytokines such as IL-6, IFN-γ, IL-4, TNF-α, IL-15, and IL-1β. These cytokines were strongly associated with a higher mortality risk, suggesting that hypoalbuminemia can serve as an indicator of poor prognosis in SARS-CoV-2 infection. These mechanisms, driven by cytokine storm and endothelial dysfunction, help explain the strong association between hypoalbuminemia and mortality risk in COVID-19. A meta-analysis of four studies found that hypoalbuminemia is associated with a higher risk of severe COVID-19 outcomes, including respiratory distress, ICU admission, and death [32]. Similarly, a retrospective cohort study by Huang et al. showed that serum albumin levels below 3.5 g/dL increased mortality risk six-fold in COVID-19 patients [33]. Our study also observed a significant decline in albumin levels from admission to outcome, underscoring the critical role of hypoalbuminemia in disease progression. Hypoalbuminemia during severe illness is often driven by inflammation, as inflammatory mediators suppress albumin synthesis in favor of producing other acute-phase proteins [34]. IL-15 can promote TNF-α release in macrophages, which induces endothelial apoptosis, vascular permeability, and capillary leak syndrome—all implicated in COVID-19 progression [35,36]. The IL-15-to-albumin ratio enhanced mortality prediction in hospitalized COVID-19 patients [37]. All these associations highlight the role of hypoalbuminemia in CRS and its potential use as a prognostic marker.

Cytokines released during inflammation are known to regulate the gene encoding cystatin C, which has been proposed to play a prognostic role in inflammatory diseases [38,39]. Cystatin C also influences the release of key inflammatory mediators such as TNF-α, IL-12, IL-10, and nitric oxide, contributing to inflammation and potential multi-organ failure in severe COVID-19 cases [40]. A systematic review and meta-analysis of 25 studies demonstrated that CysC is a valuable marker for predicting severe COVID-19, AKI, and mortality. Their finding suggests that CysC not only indicates AKI but may also highlight other renal complications, emphasizing the importance of long-term patient monitoring even after discharge [41]. In our study, we found that CysC levels were notably higher in patients who died, especially before their outcomes. At the same time, those discharged experienced a reduction in CysC levels, indicating that a decrease in CysC is likely crucial for a favorable prognosis. This aligns with the results of Chen et al., 2022, who also found a positive correlation between CysC and cytokines, including IL-8, TNF-α, and IL-6, linking inflammation to kidney injury in COVID-19 patients [42]. Similarly, our study revealed an association between CysC and cytokines like IL-4, IL-15, TNF-α, and IL-1β, further implicating the role of CRS in kidney injury and its contribution to a higher mortality risk. Consistent with Chen et al., our findings showed that increased CysC levels correlated with elevated IL-6, IL-10, IL-4, TNF-α, and IL-1β, which were all associated with a heightened mortality risk in the studied population. This reinforces that CRS significantly contributes to kidney injury and overall disease severity in SARS-CoV-2 infection.

NGAL has been implicated in several pro-inflammatory and metabolic processes in animal models of conditions such as type 2 diabetes and non-alcoholic steatohepatitis [43,44]. In COVID-19, elevated NGAL levels have been reported even in asymptomatic cases [45] and linked to complications including kidney injury, need for dialysis, shock, prolonged hospitalization, and in-hospital mortality [46]. Multiple studies have confirmed NGAL’s association with disease severity, AKI, and mortality [47,48,49,50]. Interestingly, plasma NGAL levels appear similarly elevated in patients who die with or without AKI, suggesting NGAL reflects systemic disease severity rather than kidney injury specifically [51]. Our study found significantly higher NGAL levels in patients who died compared to those discharged, both at admission and at outcome. However, the limited number of outcome samples substantially reduced statistical power. Although NGAL levels increased significantly at outcome among non-survivors, this association did not persist in the multivariable model. Therefore, these findings should be considered exploratory and warrant validation in larger, independent cohorts.

This study presents some limitations that should be acknowledged. First, the analysis was limited to two discrete time points (hospital admission and clinical outcome), which restricts the evaluation of biomarker trajectories over time and limits insights into disease progression. Although comparing values between admission and outcome provides insight into disease severity and prognosis, these data do not capture longitudinal trends. Second, kidney injury biomarkers were analyzed across the whole cohort, not only in patients with clinically diagnosed AKI, based on evidence that subclinical renal injury may occur in COVID-19 due to systemic inflammation [52]. This broader approach aimed to identify early prognostic markers relevant to a broader hospitalized population. Lastly, general severity markers such as LDH and AST were not included, limiting the completeness of risk stratification. Future studies should integrate renal, inflammatory, and systemic markers for a more comprehensive assessment.

Thus, while our findings support the prognostic value of renal biomarkers such as albumin, CysC, and NGAL, it is important to interpret these associations within their potential clinical relevance. For example, although statistically significant, the observed changes in NGAL and albumin levels were relatively modest and may not represent actionable thresholds in clinical practice. Nonetheless, the consistent inverse correlation between albumin and pro-inflammatory cytokines, alongside its decline over time, reinforces its role as a marker of systemic illness and poor prognosis in COVID-19. Similarly, the associations between CysC and inflammatory mediators suggest its utility in capturing the interplay between renal dysfunction and cytokine release syndrome (CRS). Taken together, these biomarkers may still contribute to risk stratification, particularly when integrated with other clinical and laboratory parameters. Their incorporation into composite clinical scores, rather than a reliance on isolated measurements, may offer more robust tools for identifying patients at elevated risk, guiding decisions regarding closer monitoring, early renal support, or ICU transfer. Future studies should prioritize defining clinically meaningful cutoffs and validating their predictive performance across diverse cohorts to ensure their practical application in managing patients with COVID-19 and other systemic inflammatory conditions.

## 4. Materials and Methods

### 4.1. Study Design

Patients aged 18 and older, admitted to Marcelino Champagnat Hospital with a positive real-time reverse transcriptase-polymerase chain reaction (rRT-PCR) test for SARS-CoV-2, were enrolled. After admission, the hospital conducted the second rRT-PCR to confirm the infection. Immunocompromised patients (HIV-positive or on immunosuppressive drugs), those with other viral infections, a history of solid organ or hematological transplantation, or those using tocilizumab were excluded. Out of the 446 patients initially enrolled, 45 were excluded for using tocilizumab, 6 were excluded due to unconfirmed RT-PCR results, and 1 was excluded because of a positive HIV status. This resulted in a final study group of 390 participants (Figure 8).

### 4.2. Data Collection

Comprehensive electronic data from patients, including epidemiological, clinical, laboratory, and outcome information, were obtained from medical records using Tasy^®^ (Phillips Healthcare, Andover, MA, USA). Specifically, we extracted details on gender, age, weight, underlying health conditions, admission to the Intensive Care Unit (ICU), and ultimate outcomes (discharge or death).

### 4.3. Samples Preparation

The study covered three periods during the peak of SARS-CoV-2 transmission in Brazil: April to October 2020 (180 patients, original strain, unvaccinated patients); November 2020 to February 2021 (92 patients, Alpha, Beta, and Gamma variants, no vaccinated patients); and May to August 2021 (118 patients, Alpha, Beta, and Gamma variants, 65 unvaccinated, 22 fully immunized with 3 doses and 31 with one dose). Samples were collected using SST II tubes for serum and EDTA tubes for plasma after the hospital admission and then every 72 h until discharge or death. The first and last blood samples were used for biomarker measurement for the present study. After centrifugation, serum and plasma were aliquoted into 1 mL Eppendorf tubes and stored at −80 °C for analysis.

### 4.4. Kidney Biomarkers and Inflammatory Assessment

Nephelometry was used to assess serum albumin, cystatin C (CysC), and plasma neutrophil gelatinase-associated lipocalin (NGAL) using the automated nephelometer BN ProSpec^®^ System (Siemens Medical Solutions USA, Malvern, PA, USA) according to the manufacturer’s instructions. 

### 4.5. Inflammatory Mediator Assessment

Cytokine detection (IL-4, IL-6, IL-10, IL-15, IL-1β, IFN-γ, and TNF-α) was performed using an ELISA kit (ImmunoTools, Friesoythe, Germany) in a non-sensitized 96-well plate according to the manufacturer’s instructions. Absorbance was quantified using the Versamax microplate reader (Molecular Devices, Sunnyvale, CA, USA).

### 4.6. Statistical Analysis

Quantitative variables were described using mean ± standard deviation (SD), medians, and ranges (minimum and maximum values). Categorical variables were expressed as frequencies and percentages. The Kolmogorov–Smirnov test assessed the normality of kidney biomarkers and cytokine distributions. The Student’s *t*-test was employed to compare kidney biomarker levels between two groups (discharge vs. death or admission vs. outcome sample). The Spearman correlation coefficient was used to examine the relationship between kidney biomarkers and cytokines. 

Associations between kidney biomarkers and mortality were analyzed using both univariate and multivariate logistic regression models. Covariates included in the multivariable models were selected based on statistically significant associations in univariate analysis (*p* < 0.05) and clinical relevance. The Wald test was used to assess the significance of variables, and odds ratios (OR) with 95% confidence intervals (CI) were reported to estimate associations. Additional logistic regression models were constructed to evaluate the joint effect of cytokines and kidney biomarkers on mortality risk, including each cytokine, its quadratic term, and the corresponding biomarker.

Potential effect modification between cytokines and biomarkers (serum albumin, NGAL, and cystatin C) was examined by including interaction terms in the models. Multicollinearity among covariates was assessed using the Variance Inflation Factor (VIF), with all VIF values below 2.0. Model performance was evaluated using the area under the receiver operating characteristic curve (AUC) to assess discrimination, and the Hosmer–Lemeshow goodness-of-fit test to determine calibration. 

Due to variability in sample availability for each biomarker, analyses were conducted using a complete-case approach. Only samples with available data for each specific parameter were included in the corresponding analyses. No imputation of missing data was performed, as we aimed to preserve the reliability of the original measurements and minimize potential bias introduced by estimation. Missing data primarily resulted from an insufficient sample volume or compromised sample quality during collection and processing. Statistical analyses were performed using IBM SPSS Statistics v.29.2 (IBM Corp, Armonk, NY, USA), Stata v.16.0 (StataCorp LLC, College Station, TX, USA), and GraphPad Prism v.10 (GraphPad Software, Boston, MA, USA).

## 5. Conclusions

The strength of this study lies in its ability to reveal significant disparities in renal biomarker levels from admission to outcome alongside their correlation with inflammatory mediators. This provides valuable insights into identifying individuals at higher risk of mortality. Nonetheless, certain limitations must be acknowledged, including the limited availability of NGAL measurements in the death cohort, the lack of data on sepsis-related deaths, and the inability to comprehensively classify AKI due to missing longitudinal urine output data. Despite these challenges, our findings underscore the critical role of kidney biomarker assessment in COVID-19 patients, offering predictive insights that may inform tailored interventions and enhanced surveillance of vulnerable groups. By elucidating the intricate link between renal impairment, inflammation, and clinical outcomes, this study supports the development of personalized therapeutic strategies to improve patient care, with meaningful implications for future pandemic preparedness.

## Figures and Tables

**Figure 1 ijms-26-06086-f001:**
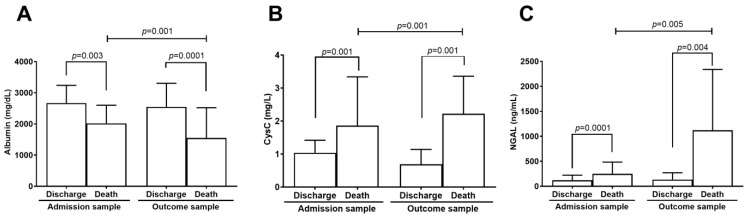
Comparison of kidney biomarker levels between discharge and death groups at admission and outcome time points. Concentrations of (**A**) albumin, (**B**) cystatin C (CysC), and (**C**) neutrophil gelatinase-associated lipocalin (NGAL) are shown as mean ± standard deviation (SD). Statistical comparisons were performed using the Student’s *t*-test, with *p*-values < 0.05 considered statistically significant. Sample sizes: Albumin: discharge group—admission (n = 304), outcome (n = 163); death group—admission (n = 55), outcome (n = 44). CysC: discharge group—admission (n = 304), outcome (n = 163); death group—admission (n = 55), outcome (n = 44). NGAL: discharge group—admission (n = 73), outcome (n = 31); death group—admission (n = 19), outcome (n = 8).

**Figure 2 ijms-26-06086-f002:**
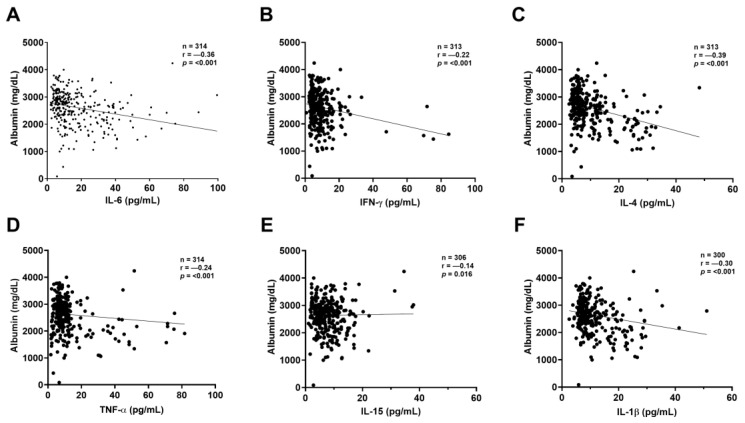
Correlation between albumin and other inflammatory mediators. Significant correlations were observed between albumin and (**A**) IL-6, (**B**) IFN-γ, (**C**) IL-4, (**D**) TNF-α, (**E**) IL-15, (**F**) IL-1β. All correlations were analyzed using the non-parametric Spearman test. Statistical significance was set at *p*-values < 0.05. Sample sizes: IL-6 (n = 337); IFN-γ: (n = 336); IL-4: (n = 334); TNF-α (n = 337); IL-15: (n = 325); IL-1β: (n = 319).

**Figure 3 ijms-26-06086-f003:**
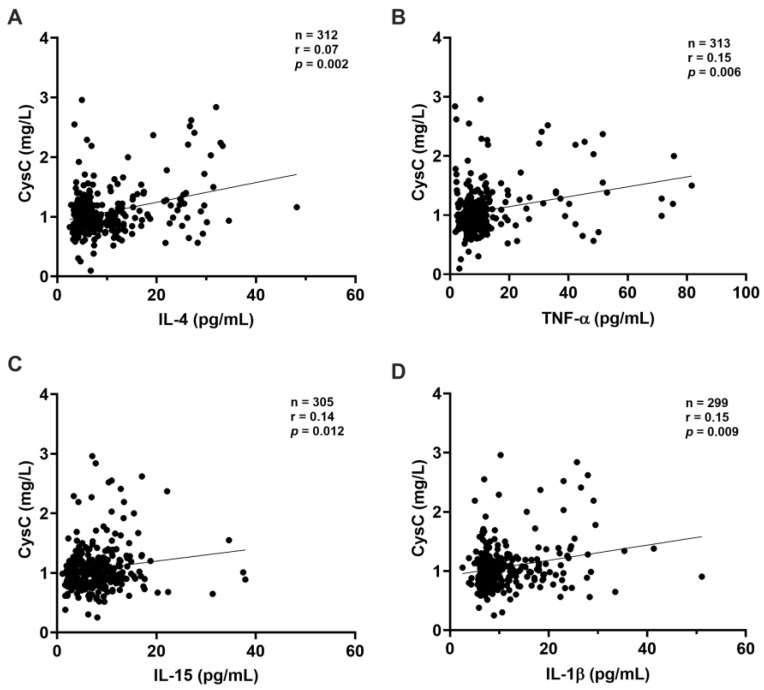
Correlation between cystatin C and other inflammatory mediators. Significant correlations were observed between albumin and (**A**) IL-4, (**B**) TNF-α, (**C**) IL-15, (**D**) IL-1β. All correlations were analyzed using the non-parametric Spearman test. Statistical significance was set at *p*-values < 0.05. Sample sizes: IL-4: (n = 334); TNF-α (n = 337); IL-15: (n = 325); IL-1β: (n = 319).

**Figure 4 ijms-26-06086-f004:**
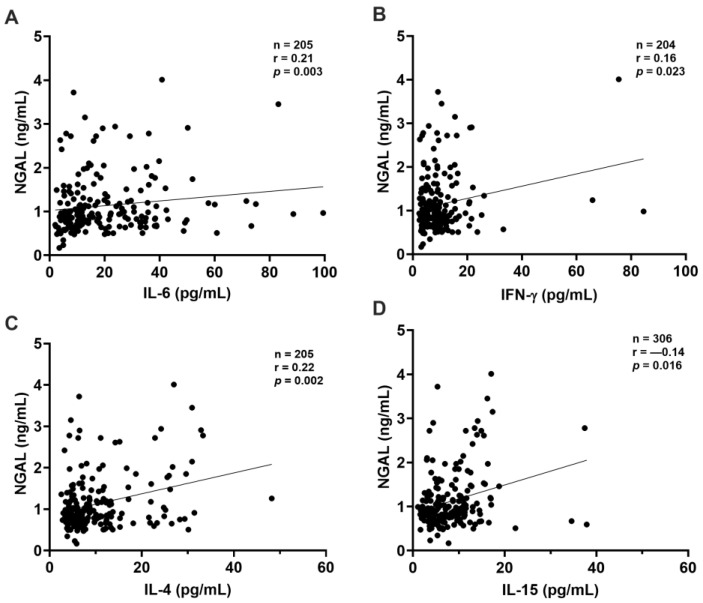
Correlation between NGAL and other inflammatory mediators. Significant correlations were observed between albumin and (**A**) IL-6, (**B**) IFN-γ, (**C**), IL-4, (**D**) IL-15. All correlations were analyzed using the non-parametric Spearman test. Statistical significance was set at *p*-values < 0.05. Sample sizes: IL-6 (n = 337); IFN-γ: (n = 336); IL-4: (n = 334); IL-15: (n = 325).

**Figure 5 ijms-26-06086-f005:**
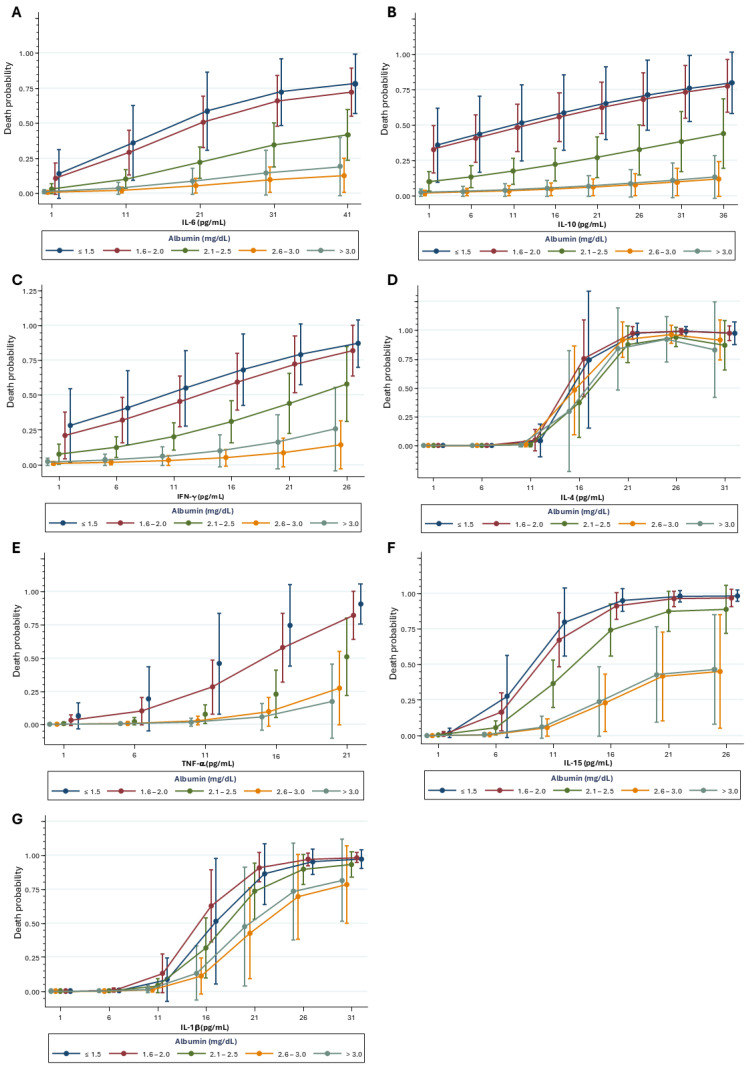
Associated effect of albumin and (**A**) IL-6, (**B**) IL-10, (**C**) IFN-γ, (**D**) IL-4, (**E**) TNF-α, (**F**) IL-15, and (**G**) IL-1β over the mortality risk. Analysis was performed using Logistic Regression with cytokine concentration, quadratic term of the cytokine, and albumin as explanatory variables. Sample sizes: IL-6 (n = 337); IL-10: (n = 324); IFN-γ: (n = 336); IL-4: (n = 334); TNF-α (n = 337); IL-15: (n = 325); IL-1β: (n = 319).

**Figure 6 ijms-26-06086-f006:**
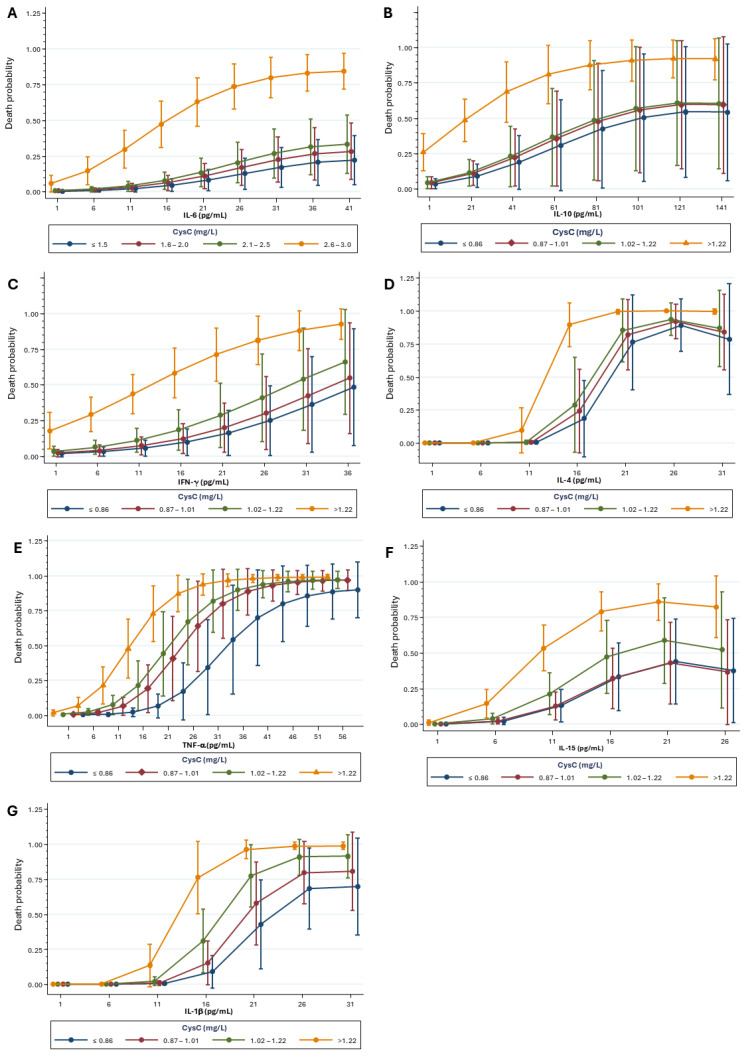
Associated effect of cystatin C and (**A**) IL-6, (**B**) IL-10, (**C**) IFN-γ, (**D**) IL-4, (**E**) TNF-α, (**F**) IL-15, (**G**) IL-1β over the mortality risk. Analysis was performed using Logistic Regression with cytokine concentration, quadratic term of the cytokine, and albumin as explanatory variables. Sample sizes: IL-6 (n = 337); IL-10: (n = 324); IFN-γ: (n = 336); IL-4: (n = 334); TNF-α (n = 337); IL-15: (n = 325); and IL-1β: (n = 319).

**Figure 7 ijms-26-06086-f007:**
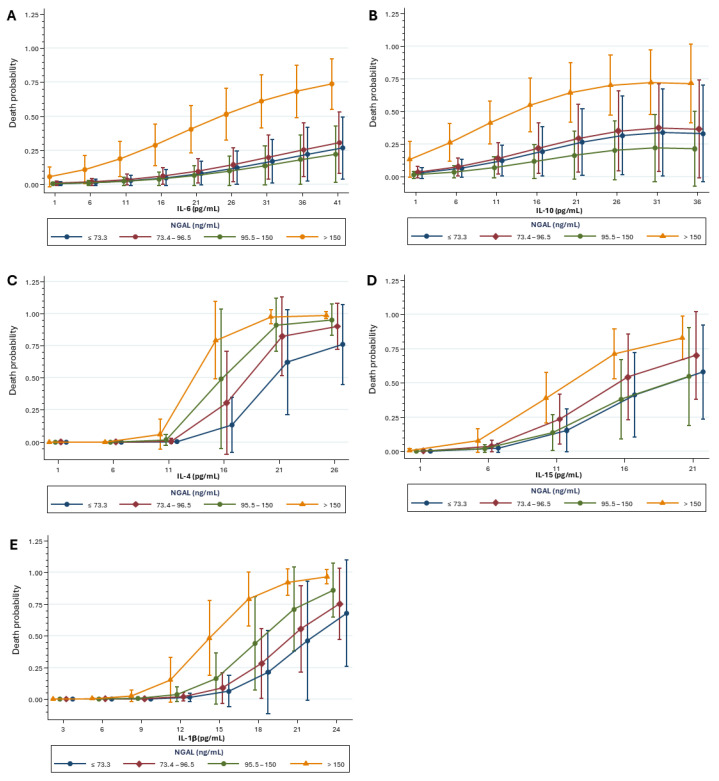
Associated effect of NGAL and (**A**) IL-6, (**B**) IL-10, (**C**) IL-4, (**D**) IL-15, (**E**) IL-1β over the mortality risk. Analysis was performed using Logistic Regression with cytokine concentration, quadratic term of the cytokine, and albumin as explanatory variables. Sample sizes: IL-6 (n = 337); IL-10: (n = 324); IL-4: (n = 334); IL-15: (n = 325); and IL-1β: (n = 319).

**Figure 8 ijms-26-06086-f008:**
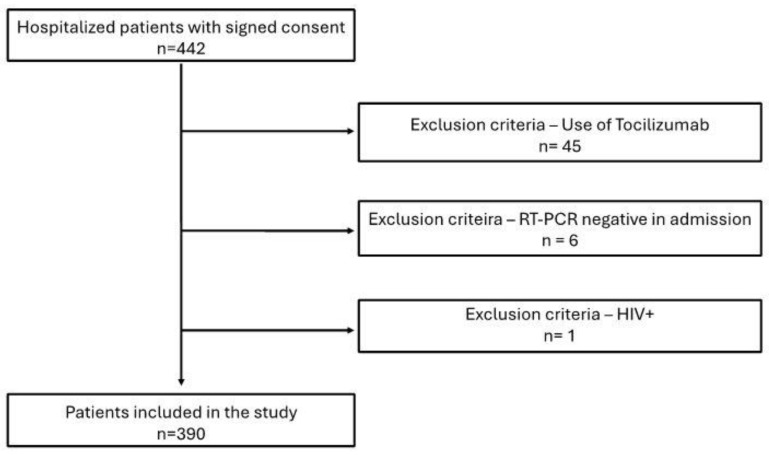
Cohort flow chart illustrating the inclusion and exclusion criteria of participants in the study.

**Table 1 ijms-26-06086-t001:** Demographic and clinical characteristics of participants.

Characteristics at Admission	n	Mean ± SD or %
Age (years)	390	54.6 ± 16.4
Sex (male)	243	62.3%
Creatinine (mg/dL)	358	10 ± 0.6
GFR (mL/min/1.73 m^2^)	359	76 ± 30
SpO_2_	389	91.6 ± 5.3
BMI (kg/m^2^)	342	29.9 ± 0.9
Comorbidities at admission		
SAH	154	39.5%
DM	89	22.8%
Kidney disease ^1^	13	3.3%
COPD	12	3.1%
Cardio-metabolic disease ^2^	200	52.2%
During hospitalization		
Need of hemodialysis	20	5.13%
Renal complications ^3^	40	10.25%

GFR, glomerular filtration rate. SpO_2_, peripheral oxygen saturation. BMI, body mass index. SAH, systemic arterial hypertension. DM, diabetes mellitus. ^1^ Includes chronic kidney disease, nephrectomy, and transplant. COPD, chronic obstructive pulmonary disease. ^2^ Includes SAH, DM, cardiac insufficiency, chronic coronary syndrome, acute myocardial infarction, brain stroke, peripheral arterial disease, and dyslipidemia. ^3^ Includes acute kidney injury; acute-on-chronic kidney disease.

**Table 2 ijms-26-06086-t002:** Association between demographic and clinical characteristics and death.

Demographic and Clinical Variables	Classification	Total	n (%) Death	^1^*p* Value	OR (IC 95%)
Age (years)	<65	285	19 (6.7%)	<0.001	8.97 (4.88–16.48)
	≥65	105	41 (39%)		
Cardio-metabolic disease ^2^	No	183	11 (6%)	<0.001	4.8 (2.41–9.59)
	Yes	200	47 (23.5%)		
Number of comorbidities	<4	321	34 (10.6%)	<0.001	5.1 (2.79–9.33)
	≥4	69	26 (37.7%)		
Days of symptoms before admission	>7	208	20 (9.6%)	0.001	2.65 (1.48–4.73)
	≤7	182	40 (22%)		
SpO_2_	≥95%	132	12 (9.1%)	0.015	2.3 (1.17–4.49)
	<95%	257	48 (18.7%)		
ICU at admission	No	308	25 (8.1%)	<0.001	8.8 (4.82–16.07)
	Yes	80	35 (43.8%)		
Severity (WHO)	Mild	122	14 (11.5%)	0.152	1.6 (0.84–3.04)
	Severe	262	45 (17.2%)		

^1^ Logistic Regression Univariate Model and Wald test, *p* < 0.05. ^2^ Includes SAH, DM, cardiac insufficiency, chronic coronary syndrome, acute myocardial infarction, brain stroke, peripheral arterial disease, and dyslipidemia. SpO_2_, peripheral oxygen saturation. ICU, Intensive care unit. WHO, World Health Organization.

**Table 3 ijms-26-06086-t003:** Association of the kidney biomarker levels and death.

Biomarker	Outcome	n	Mean ± SD	Median (Min–Max)	^1^*p* (uni)	OR (IC 95%)	^2^*p* (multi)	OR (IC 95%)
			Admission sample					
Albumin (mg/dL)	Discharge	304	2668 ± 569	2740 (85–4000)	<0.001	0.17 (0.10–0.30)	<0.001	0.25 (0.13–0.49)
	Death	55	2019 ± 581	2010 (1060–4240)				
CysC (mg/L)	Discharge	304	1.04 ± 0.38	1 (0.10–3.93)	<0.001	6.78 (3.63–12.7)	0.011	2.74 (1.26–5.94)
	Death	55	1.86 ± 1.48	1.4 (0.48–9.91)				
NGAL (ng/mL)	Discharge	196	118.9 ± 103.1	91.7 (15.6–955)	<0.001	2.91 (1.85–4.56)	0.009	2.11 (1.21–3.68)
	Death	44	247.8 ± 236.2	179 (53.5–1300)				
			Outcome sample					
Albumin (mg/dL)	Discharge	163	2545 ± 759	2700 (115–5920)	<0.001	0.12 (0.06–0.23)	<0.001	0.14 (0.07–0.29)
	Death	44	1549 ± 971	1365 (483–7030)				
CysC (mg/L)	Discharge	163	0.96 ± 0.45	0.90 (0.01–4.87)	<0.001	13.9 (5.47–35.4)	<0.001	12.5 (4.58–34.0)
	Death	44	2.22 ± 1.14	1.82 (0.71–4.95)				
NGAL (ng/mL)	Discharge	31	133.2 ± 137	94.2 (47.8–722)	0.040	1.81 (1.03–3.20)	<0.001	-
	Death	8	1121 ± 1217	414 (76.1–2990)				

^1^. Univariate Logistic Regression Model and Wald test, *p* < 0.05. ^2^. Multivariate Logistic Regression Model and Wald test, *p* < 0.05. Adjusted for age (<65 or ≥65), cardiometabolic disease (SAH, DM, cardiac insufficiency, chronic coronary syndrome, acute myocardial infarction, brain stroke, peripheral arterial disease, and dyslipidemia), SpO_2_ at admission (<95% or ≥95%), comorbidities (<4 or ≥4), days of symptoms before admission (<7 or ≥7), and ICU at admission. Multicollinearity was assessed using the Variance Inflation Factor (VIF), and all variables included in the model presented VIF values < 2 (maximum VIF = 1.88), indicating no evidence of problematic collinearity. CysC, Cystatin C; NGAL, Plasmatic neutrophil gelatinase-associated lipocalin.

## Data Availability

The data supporting this study’s findings are available from the corresponding author (Andréa Novais Moreno-Amaral) upon reasonable request.

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
