# Peer review of "The Relationship Between Kidney Biomarkers, Inflammation, Severity, and Mortality Due to COVID-19—A Two-Timepoint Study"

_ijms, 2025, doi:10.3390/ijms26136086_

Round 1
Reviewer 1 Report
Comments and Suggestions for Authors
Title of the Article: Temporal Dynamics of Kidney Biomarkers and Inflammatory Mediators in SARS-CoV-2 Progression: Implications for Disease Severity and Mortality Risk
General Comments:
The impact of COVID-19 disease severity and mortality on inflammatory mediators’ and kidney injury biomarkers’ interactions is the subject of this manuscript. It adds to the literature by including a large patient cohort and applying relevant biomarkers. Nevertheless, several important gaps undermine the rigor and scope of the interpretations. Among these gaps are the lack of reporting adequate statistical detail, vague explanations of the “temporal dynamics” phrasing, restrictions regarding the sampling of the biomarkers, and excessive conclusions of the results. A complete revision is required to balance scientific integrity and trustworthiness with precision and transparency
Major Scientific Concerns:
Misleading Title:
The title suggests a long-term study, yet only two points in time were evaluated—admission and outcome. The phrase “temporal dynamics” in this regard is misleading.
Recommendation: Consider using a more precise title like “The Relationship Between Kidney Biomarkers, Inflammation, Severity, and Mortality of COVID-19—A Two-Timepoint Study.”
Confounder Adjustment Insufficient:
As some variables were already controlled for in the multivariable models, there remains critique on the failure to include model performance metrics (R², AUC) and not addressing multicollinearity or rationalizing selected covariates.
Suggestion: Add how the potential confounding interactions among the cytokines and the biomarkers were dealt with.
Limited Temporal Interpretation:
The analysis provides two-time cross-sectional analyses which does not justify either “progression” or “dynamics” as reasoning.
Suggestion: Discuss this limitation as a two-timepoint analysis and not longitudinal..
Small Sample Size for NGAL:
NGAL was assessed in only 31 patients for the discharge group and 8 for the death group at outcome timepoint. This is a major concern because it considerably undermines the statistical power and dependability of the conclusions pertaining to NGAL.
Recommendation: The authors should consider stating this explicitly and, at the very least, temper any conclusions involving NGAL.
No Stratification for Vaccination:
While the manuscript records differences in vaccination status by time period, none of the analyses treat vaccination as a covariate. This is an analysis flaw given that vaccination impacts COVID-19 outcomes.
Suggestion: Provide analysis restricted to particular subgroups or account for vaccination status.
Poor Table Formatting:
In Table 1, column headers such as “Title 2” and “Title 3” are unclear and unprofessional.
Suggestion: Replace them with informative labels.
Figures Lack Sample Size Clarity:
Sampling numbers per group are not uniform based on the provided figures.
Recommendation: Include sample sizes (n) in the figure legends.
Statistical vs Clinical Significance:
Even though many differences may achieve statistical significance, in cases such as small alterations in NGAL or albumin, the clinical relevance is neglected.
Interpret the findings from the perspective of their possible clinical effect.
Methodological Concerns:
Cytokine Panel Not Justified:
No rationale is given for adding IL-4 and IL-15 given that they are not normally integral components of COVID-19 pathophysiology.
Recommendation: Provide justification for the chosen markers or omit irrelevant ones.
Handling of Missing Data Not Addressed:
The sample size variation across parameters (for instance, albumin compared to NGAL) indicates potential missing values, which the manuscript fails to address.
Recommendation: Explain your strategy, such as whether you chose complete-case analysis, imputation, or exclusion.
Discussion and Interpretation Issues:
Overstatement of Novelty:
The discussion states that there is some novelty in the findings which simply confirm previous research, such as albumin and CysC being predictors of mortality.
Recommendation: Acknowledge the confirmatory nature of the findings.
Missing of Biological Insight:
There is a lack of biological reasoning for links connecting cytokines with kidney biomarkers.
Recommendation: Refer relevant experimental data or expand the discussion with mechanistic details.
There are some grammatical errors and backward constructions that affect clarity. Try to reduce these mistakes for more clarity.
Examples:
-
“The death group showed significantly lower levels in the outcome sample.”
-
“Correlation between albumin and IL-10 (data not shown).
Author Response
Comments from Reviewer 1:
We thank the reviewer for the thorough and constructive feedback provided. All comments have been addressed point by point below. Corresponding changes have been made in the revised manuscript and are highlighted in red for ease of reference.
General Comments:
The impact of COVID-19 disease severity and mortality on inflammatory mediators’ and kidney injury biomarkers’ interactions is the subject of this manuscript. It adds to the literature by including a large patient cohort and applying relevant biomarkers. Nevertheless, several important gaps undermine the rigor and scope of the interpretations. Among these gaps are the lack of reporting adequate statistical detail, vague explanations of the “temporal dynamics” phrasing, restrictions regarding the sampling of the biomarkers, and excessive conclusions of the results. A complete revision is required to balance scientific integrity and trustworthiness with precision and transparency.
Major Scientific Concerns:
Misleading Title: The title suggests a long-term study, yet only two points in time were evaluated—admission and outcome. The phrase “temporal dynamics” in this regard is misleading.
Recommendation: Consider using a more precise title like “The Relationship Between Kidney Biomarkers, Inflammation, Severity, and Mortality of COVID-19—A Two-Timepoint Study.”
Response: We thank the reviewer for this insightful comment. We agree that the original title may imply a longer follow-up or multiple timepoint assessments, which could be misleading given that our analysis focuses on two specific timepoints: hospital admission and clinical outcome. To address this, we agree with the reviewer’s observation and have accepted the suggested title revision to reflect the study’s design and scope more accurately.
Confounder Adjustment Insufficient: (1) As some variables were already controlled for in the multivariable models, there remains critique on the failure to include model performance metrics (R², AUC) (2) and not addressing multicollinearity or rationalizing selected covariates.
Suggestion: Add how the potential confounding interactions among the cytokines and the biomarkers were dealt with.
Response: (1) We appreciate the reviewer’s suggestion regarding including model performance metrics. All multivariable models in our study were logistic regressions. To evaluate their performance, we calculated the area under the ROC curve (AUC) to assess the model's discriminative ability, where 1.0 indicates excellent discrimination (Table below). In addition, we conducted the Hosmer–Lemeshow goodness-of-fit test to evaluate model calibration. A non-significant result (p>0.05) suggests a good fit (Table below). These performance metrics are now reported in the revised manuscript, in the Methods section, and support the adequacy of the models used.
Table - Summary of model performance
Model |
AUC (95% CI) |
Hosmer-Lemeshow test p-value |
Table 3 - Admission |
|
|
Albumin (mg/dL) |
0.90 (0.85 – 0.94) |
0.277 |
CysC (mg/L) |
0.85 (0.79 – 0.91) |
0.244 |
NGAL (ng/mL) |
0.84 (0.77 – 0.91) |
0.652 |
Table 3 - Outcome |
|
|
Albumin (mg/dL) |
0.91 (0.87 – 0.96) |
0.764 |
CysC (mg/L) |
0.91 (0.87 – 0.96) |
0.149 |
Figure 5 models |
|
|
A |
0.86 (0.79 – 0.92) |
0.277 |
B |
0.84 (0.78 – 0.91) |
0.353 |
C |
0.88 (0.83 – 0.93) |
0.865 |
D |
0.94 (0.90 – 0.99) |
0.868 |
E |
0.95 (0.90 – 0.99) |
0.143 |
F |
0.92 (0.88 – 0.96) |
0.133 |
G |
0.98 (0.96 – 0.99) |
0.882 |
Figure 6 models |
|
|
A |
0.86 (0.80 – 0.92) |
0.412 |
B |
0.80 (0.71 – 0.88) |
0.248 |
C |
0.85 (0.78 – 0.92) |
0.908 |
D |
0.90 (0.83 – 0.97) |
0.994 |
E |
0.90 (0.82 – 0.96) |
0.089 |
F |
0.88 (0.83 – 0.93) |
0.758 |
G |
0.98 (0.96 – 0.99) |
0.512 |
Figure 7 models |
|
|
A |
0.82 (0.73 – 0.92) |
0.070 |
B |
0.78 (0.68 – 0.88) |
0.092 |
C |
0.98 (0.97 – 0.99) |
0.747 |
D |
0.89 (0.83 – 0.94) |
0.847 |
E |
0.97 (0.95 – 0.99) |
0.259 |
(2) Regarding the point of not addressing multicollinearity or rationalizing selected covariates: all covariates included in the multivariable models (age, cardiometabolic disease, SpO₂ at admission, number of comorbidities, days of symptoms before admission, and ICU admission) were selected based on their statistically significant association with mortality in the univariate analyses (p < 0.05), as well as their clinical relevance as potential confounders. To address concerns about multicollinearity, we conducted a Variance Inflation Factor (VIF) analysis for all covariates included in the models. All VIF values were below the conventional threshold of concern (VIF < 5), with the highest being 1.88, indicating no evidence of problematic collinearity. This rationale has now been clarified in the Methods section and the legend of Table 3.
Addressing the valuable suggestion from the reviewer regarding potential confounding interactions between the cytokines and clinical biomarkers: we conducted an exploratory analysis to assess effect modification between cytokines (IL-6, IL-10, IFN-γ, IL-4, TNF-α, IL-15, and IL-1β) and key clinical biomarkers (serum albumin, NGAL, and cystatin C). To this end, we tested interaction terms in the multivariable logistic regression models in Figures 5, 6, and 7. Each model included the cytokine (entered as linear and quadratic terms to capture potential non-linear associations), the clinical biomarker, and their corresponding interaction terms. None of the interaction terms reached statistical significance at the 0.05 level, suggesting no evidence of meaningful effect modification between the cytokines and the biomarkers on the outcome. The revised Methods section clearly describes this analytical strategy.
Limited Temporal Interpretation: The analysis provides two-time cross-sectional analyses which does not justify either “progression” or “dynamics” as reasoning.
Suggestion: Discuss this limitation as a two-timepoint analysis and not longitudinal.
Response: We thank the reviewer for this important observation. We agree that the use of terms such as “progression” or “dynamics” could be misinterpreted as suggesting a longitudinal study design, which does not reflect our two-time-point (admission and outcome) analysis. In response, a new paragraph with the study's limitations was added in the Discussion section to further contextualize this and other methodological considerations. We also carefully reviewed and revised the entire manuscript to remove or rephrase all terms that could imply a longitudinal approach.
Small Sample Size for NGAL: NGAL was assessed in only 31 patients for the discharge group and 8 for the death group at outcome timepoint. This is a major concern because it considerably undermines the statistical power and dependability of the conclusions pertaining to NGAL.
Recommendation: The authors should consider stating this explicitly and, at the very least, temper any conclusions involving NGAL.
Response: We thank the reviewer for highlighting this critical limitation. We fully agree that the small number of NGAL measurements at the outcome time point, particularly in the death group, limits the statistical power and generalizability of the conclusions drawn for this biomarker. We have revised the manuscript in the Results section to address this point and in the Discussion section to acknowledge this limitation. Furthermore, we have tempered the interpretation of the findings related to NGAL by presenting them as preliminary observations that require confirmation in larger cohorts.
No Stratification for Vaccination: While the manuscript records differences in vaccination status by time period, none of the analyses treat vaccination as a covariate. This is an analysis flaw given that vaccination impacts COVID-19 outcomes.
Suggestion: Provide analysis restricted to particular subgroups or account for vaccination status.
Response: We thank the reviewer for this thoughtful and important observation. We fully acknowledge the relevance of vaccination status in shaping COVID-19 outcomes. However, due to the timing of our cohort, we faced significant limitations in capturing this information. Vaccination in Curitiba, Brazil, began in January 2021, but was initially restricted to healthcare professionals. The broader rollout to elderly individuals and high-risk groups only started in April 2021. Our data collection period ended in August 2021, meaning most participants were enrolled before large-scale vaccination was available. As a result, of the 118 patients included during the large-scale vaccination in our city, only 53 individuals in our cohort had recorded vaccination status; among them, only 22 were fully vaccinated according to the recommended immunization schedule. Given this extremely limited and non-representative data, we judged that including vaccination status as a covariate in multivariable models would compromise the stability and interpretability of the analyses. Therefore, vaccination was not incorporated into the final models. We fully agree that immunization plays a critical role in COVID-19 outcomes in the current phase of the pandemic. However, our cohort largely reflects a pre-vaccination context, and the associations observed between kidney biomarkers, cytokines, and mortality should be interpreted within that framework. We have now clarified this point in the revised Discussion section. We also agree that future studies should investigate how vaccination status may interact with or modify the prognostic value of these biomarkers in more recent, post-vaccination cohorts.
Poor Table Formatting: In Table 1, column headers such as “Title 2” and “Title 3” are unclear and unprofessional.
Suggestion: Replace them with informative labels.
Response: We thank the reviewer for pointing out the issue with the column headers in Table 1. We apologize for the oversight during the table's formatting and acknowledge that using placeholder titles such as “Title 2” and “Title 3” was unintentional and unprofessional. In the revised version, we corrected the nomenclature and replaced these placeholders with accurate and descriptive labels representing the data shown. We believe this change significantly improves the clarity and presentation of the table.
Figures Lack Sample Size Clarity: Sampling numbers per group are not uniform based on the provided figures.
Recommendation: Include sample sizes (n) in the figure legends.
Changes have been made in the figure legends throughout the revised manuscript.
Response: We appreciate the reviewer’s observation regarding the lack of clarity in sample sizes across the figures. We agree that indicating the number of samples per group is essential for transparency and proper interpretation of the results. In response, we have updated all legends to include each group's corresponding sample sizes (n).
Statistical vs Clinical Significance: Even though many differences may achieve statistical significance, in cases such as small alterations in NGAL or albumin, the clinical relevance is neglected.
Interpret the findings from the perspective of their possible clinical effect.
Response: We thank the reviewer for this important observation. We agree that statistical significance does not always imply clinical relevance, particularly when dealing with changes in biomarkers such as albumin or NGAL, due to the limited number of samples. In response, we have added a paragraph to the Discussion section to address the interpretation of our findings in the context of their potential clinical impact. Specifically, we now highlight which alterations, despite being statistically significant, may have limited clinical utility, and emphasize findings that may hold greater promise for guiding patient management or risk stratification.
Methodological Concerns:
Cytokine Panel Not Justified: No rationale is given for adding IL-4 and IL-15 given that they are not normally integral components of COVID-19 pathophysiology.
Recommendation: Provide justification for the chosen markers or omit irrelevant ones.
Response: We appreciate the reviewer’s comment regarding including IL-4 and IL-15 in our cytokine panel. While we understand the concern that these interleukins are not always considered central to COVID-19 pathophysiology, we chose them based on a recent publication of our group showing a prospective variation of inflammatory mediators in SARS-CoV-2 infection. We have included a paragraph in both the Introduction and Discussion sections to address this comment and provide appropriate context.
Handling of Missing Data Not Addressed:
The sample size variation across parameters (for instance, albumin compared to NGAL) indicates potential missing values, which the manuscript fails to address.
Recommendation: Explain your strategy, such as whether you chose complete-case analysis, imputation, or exclusion.
Response: We thank the reviewer for this valuable comment. Indeed, variations in sample size across parameters such as albumin and NGAL are due to missing data, primarily resulting from limitations in sample volume and quality during collection and processing. We acknowledge that this aspect was not addressed in the initial version of the manuscript. In this study, we employed a complete-case analysis approach, where only available data for each parameter were included in the respective analyses. No data imputation was performed. This strategy was chosen to preserve the integrity of the original measurements and avoid introducing bias through estimation. We have now included a statement clarifying this approach in the revised version of the Methods section.
Discussion and Interpretation Issues:
Overstatement of Novelty:
The discussion states that there is some novelty in the findings which simply confirm previous research, such as albumin and CysC being predictors of mortality.
Recommendation: Acknowledge the confirmatory nature of the findings.
Response: We appreciate the reviewer’s observation regarding the interpretation of novelty in our findings. We agree that the associations of albumin and cystatin C with mortality have been reported in previous studies, and we acknowledge that our results partially confirm and reinforce these prior findings. In the revised Discussion section, we have tempered the language to reflect the confirmatory nature of these observations more accurately. At the same time, we emphasize that validating these associations in our specific cohort adds valuable context and supports the broader generalizability of these biomarkers in predicting COVID-19 outcomes.
Missing of Biological Insight:
There is a lack of biological reasoning for links connecting cytokines with kidney biomarkers.
Recommendation: Refer relevant experimental data or expand the discussion with mechanistic details.
Response: We thank the reviewer for this insightful comment. We agree that a more precise biological rationale linking cytokines to kidney biomarkers would strengthen the interpretation of our findings. In the revised Discussion section, we have expanded on the potential mechanistic pathways supported by experimental data and literature. Additionally, we highlight how cytokine-driven immune activation may exacerbate renal inflammation. These additions provide a more comprehensive biological framework to support the observed correlations and strengthen the translational relevance of our findings.
Comments on the Quality of English Language
There are some grammatical errors and backward constructions that affect clarity. Try to reduce these mistakes for more clarity.
Examples:
“The death group showed significantly lower levels in the outcome sample.”
“Correlation between albumin and IL-10 (data not shown).
Response: We thank the reviewer for pointing out the issues related to language clarity and grammar. We have carefully reviewed the manuscript and revised sentences with grammatical errors and unclear constructions to improve overall readability. The revised manuscript has been thoroughly edited to ensure proper grammar, syntax, and flow throughout.
Reviewer 2 Report
Comments and Suggestions for Authors
The authors have studied the relationship of kidney biomarkers CysC and NGAL as well as low albumin concentrations with the outcome of a severe COVID-19 infection. The findings of this study are important as outcome, ie survival or death is an important factor to predict for COVID-19 patients. The authors should compare their findings with those of other authors as well as with the general discussion in the literature regarding outcome prediction in COVID-19 disease. The authors should add a paragraph in the discussion which describes what are the consequences of their findings for clinical practice in the management of COVID-19 disease.
Author Response
Comments from Reviewer 2:
We thank the reviewer for the thorough and constructive feedback provided. All comments have been addressed point by point below. Corresponding changes have been made in the revised manuscript and are highlighted in red for ease of reference.
The authors have studied the relationship of kidney biomarkers CysC and NGAL as well as low albumin concentrations with the outcome of a severe COVID-19 infection. The findings of this study are important as outcome, ie survival or death is an important factor to predict for COVID-19 patients.
The authors should compare their findings with those of other authors as well as with the general discussion in the literature regarding outcome prediction in COVID-19 disease.
The authors should add a paragraph in the discussion which describes what are the consequences of their findings for clinical practice in the management of COVID-19 disease.
Response: We thank the reviewer for their valuable feedback and for recognizing the importance of our study in predicting outcomes in severe COVID-19 cases. In response to the suggestion, we have revised the Discussion section to include a comparative analysis of our findings with those reported in the current literature. Specifically, we have discussed how our results regarding elevated NGAL and CysC levels and decreased albumin concentrations align with or differ from previously published studies on prognostic markers in COVID-19.
Additionally, as requested, we have added a final paragraph in the Discussion section to discuss the potential clinical implications of our findings. This paragraph emphasizes how the identified biomarkers may support early risk stratification and guide clinical decision-making for severely ill COVID-19 patients, particularly in settings with limited resources.
We believe these additions have strengthened our work's clinical relevance and contextualization within the broader scientific literature.
Reviewer 3 Report
Comments and Suggestions for Authors
Sara Soares Tozoni et al. determined the levels of kidney injury biomarkers and inflammatory mediators in COVID-19 patients' blood samples to reveal the correlation of these markers with disease severity. The results demonstrated that these markers strongly correlated with mortality in COVID-19 patients, highlighting their potential application in identifying high-risk patients. The major issue for this study is its rationale. According to the study population, only 40 patients (10.25%) had renal complications. However, the authors assessed kidney injury biomarkers for the entire study population. In contrast, 257 patients had low SpO2 levels. This is similar to the high severity patient group, underscoring the importance of determining general markers like LDH, AST, et al. for patients. I was wondering if the authors collected data on these general markers. A minor point is that the authors need to specify 'title 1' and 'title 2' in Table 1.
Author Response
Comments from Reviewer 3:
We thank the reviewer for the thorough and constructive feedback provided. All comments have been addressed point by point below. Corresponding changes have been made in the revised manuscript and are highlighted in red for ease of reference.
Sara Soares Tozoni et al. determined the levels of kidney injury biomarkers and inflammatory mediators in COVID-19 patients' blood samples to reveal the correlation of these markers with disease severity. The results demonstrated that these markers strongly correlated with mortality in COVID-19 patients, highlighting their potential application in identifying high-risk patients.
The major issue for this study is its rationale. According to the study population, only 40 patients (10.25%) had renal complications. However, the authors assessed kidney injury biomarkers for the entire study population.
In contrast, 257 patients had low SpO2 levels. This is similar to the high severity patient group, underscoring the importance of determining general markers like LDH, AST, et al. for patients. I was wondering if the authors collected data on these general markers.
A minor point is that the authors need to specify 'title 1' and 'title 2' in Table 1.
Response: We thank the reviewer for their thoughtful evaluation and constructive feedback.
Major Point – Study Rationale and Population Selection:
We appreciate the reviewer’s concern regarding the rationale for evaluating kidney injury biomarkers in the entire cohort, despite only a subset of patients (10.25%) presenting with documented renal complications. Our reasoning was based on accumulating evidence that subclinical kidney injury may occur in COVID-19 patients, even in the absence of overt renal dysfunction, and that early alterations in renal biomarkers can provide prognostic insights. Therefore, we aimed to explore whether such biomarkers could serve as early indicators of poor outcomes across a broader patient population, not limited to those with clinically apparent kidney involvement.
Regarding General Severity Markers (LDH, AST, etc.):
We agree with the reviewer that general markers of tissue damage and systemic inflammation, such as LDH and AST, are essential for assessing disease severity in COVID-19. Unfortunately, these parameters were not measured in our cohort due to limitations in sample volume and the prioritization of cytokine and kidney injury panels during laboratory processing. We recognize that the absence of these markers limits the comprehensiveness of our analysis and may impact the depth of risk stratification. This limitation has been explicitly acknowledged and discussed in the revised Discussion section as a methodological constraint and a consideration for future studies aiming to integrate renal, inflammatory, and general severity biomarkers.
Minor Point – Table 1 Clarification:
We thank the reviewer for noting the labeling issue. The headings "title 1" and "title 2" in Table 1 were placeholders and have now been replaced with appropriate and descriptive labels to improve clarity and presentation.
Round 2
Reviewer 1 Report
Comments and Suggestions for Authors
Suitable for publication
Reviewer 3 Report
Comments and Suggestions for Authors
The manuscript is much improved after revesion and ready for publication.